# Non-alcoholic Fatty Liver Disease (NAFLD), Type 2 Diabetes, and Non-viral Hepatocarcinoma: Pathophysiological Mechanisms and New Therapeutic Strategies

**DOI:** 10.3390/biomedicines11020468

**Published:** 2023-02-06

**Authors:** Erica Vetrano, Luca Rinaldi, Andrea Mormone, Chiara Giorgione, Raffaele Galiero, Alfredo Caturano, Riccardo Nevola, Raffaele Marfella, Ferdinando Carlo Sasso

**Affiliations:** Department of Advanced Medical and Surgical Sciences, Università della Campania “Luigi Vanvitelli”, 80138 Naples, Italy

**Keywords:** nonalcoholic fatty liver disease, type 2 diabetes, hepatocellular carcinoma

## Abstract

In recent years, the incidence of non-viral hepatocellular carcinoma (HCC) has increased dramatically, which is probably related to the increased prevalence of metabolic syndrome, together with obesity and type 2 diabetes mellitus (T2DM). Several epidemiological studies have established the association between T2DM and the incidence of HCC and have demonstrated the role of diabetes mellitus as an independent risk factor for the development of HCC. The pathophysiological mechanisms underlying the development of Non-alcoholic fatty liver disease (NAFLD) and its progression to Non-alcoholic steatohepatitis (NASH) and cirrhosis are various and involve pro-inflammatory agents, oxidative stress, apoptosis, adipokines, JNK-1 activation, increased IGF-1 activity, immunomodulation, and alteration of the gut microbiota. Moreover, these mechanisms are thought to play a significant role in the development of NAFLD-related hepatocellular carcinoma. Early diagnosis and the timely correction of risk factors are essential to prevent the onset of liver fibrosis and HCC. The purpose of this review is to summarize the current evidence on the association among obesity, NASH/NAFLD, T2DM, and HCC, with an emphasis on clinical impact. In addition, we will examine the main mechanisms underlying this complex relationship, and the promising strategies that have recently emerged for these diseases’ treatments.

## 1. Introduction

Metabolic syndrome (MetS) has greatly increased in prevalence, along with the exponential increase of obesity worldwide [1]. Metabolic syndrome encompasses a group of metabolic disorders such as hypertension, central obesity, insulin resistance (IR), and atherogenic dyslipidemia and is strongly associated with an increased risk of developing diabetes and atherosclerotic and non-atherosclerotic cardiovascular disease (CVD), as well as liver fibrosis and hepatocarcinoma [2]. Genetic and acquired factors both contribute to MetS’ pathogenesis, as well as to the inflammation pathway leading to CVD and liver fibrosis [3]. Hepatocellular carcinoma (HCC) is an aggressive primary liver cancer and is the third leading cause of cancer death worldwide [4]. HCC predisposing factors include liver cirrhosis, hepatitis B and C infections (HBV-HCV), non-alcoholic fatty liver disease (NAFLD), and, particularly, Type 2 diabetes mellitus (T2DM) [5]. This evidence has been confirmed by several studies, which reported an increased risk of developing HCC in T2DM patients, even in the absence of alcoholism, obesity, and chronic hepatitis [6,7]. By contrast, NAFLD plays an important role in the increased incidence of T2DM and its complications [8], through the development of IR, which is a key linking factor between these two diseases [9]. In recent years, vaccinations and newer highly effective oral antiviral therapies, including direct-acting antivirals, have reduced the risk of viral hepatocellular carcinoma [10]. However, a current analysis predicts an increase in HCC incidence driven by NAFLD, leading to a total of 12,240 cases per year by 2030 in the United States [11]. In fact, about 35% of non-alcoholic steatohepatitis (NASH) cases progress to liver fibrosis and potentially to end-stage liver disease or HCC [12,13].

NAFLD includes a large spectrum of diseases ranging from hepatic steatosis, fibrosis, NASH, to end-stage liver cirrhosis [14]. It has recently been described that up to 20% of individuals with NAFLD and T2DM will develop clinically significant liver fibrosis [15]. In the year 2020, a consensus of international experts proposed going beyond the current nomenclature “non-alcoholic fatty liver disease” (NAFLD) and change it to the acronym MAFLD, “metabolic dysfunction associated with fatty liver disease”, to further enhance the underlying condition of systemic metabolic dysfunction [16]. The diagnosis of MAFLD is based on the detection of hepatic steatosis (diagnosed by imaging, biomarkers, or histology) and at least one characteristic of overweight/obesity, T2DM, or metabolic dysregulation [17]. The last criterion is met when at least two features are present, including increased waist circumference, hypertension, hypertriglyceridemia, low HDL-C, prediabetes, IR, and inflammation or subclinical inflammation [16,18]. The MAFLD definition represents a paradigm shift that is able to identify a more homogeneous group of patients with fatty liver disease secondary to metabolic impairment. However, the impact of the new classification in clinical practice is not yet known. MAFLD patients with advanced fibrosis and cirrhosis have a higher risk of developing HCC, although HCC can occur even in the absence of advanced fibrosis [19]. Furthermore, it has been observed that patients with MAFLD-HCC have less severe liver damage and dysfunction than patients with HCV-related HCC, as indicated by higher serum albumin, lower serum bilirubin, and lower rates of ascites.

Phenotypically, MAFLD-HCCs appear to be different from other forms of HCCs, as the lesions are generally well-differentiated, solitary, with a greater inflammatory infiltration, lower likelihood of extrahepatic metastasis [20], and a greater size than HCC originating from other chronic liver diseases [21]. The purpose of this review is to examine the relationship between NAFLD, metabolic syndrome, and HCC from a pathophysiological point of view. Moreover, the therapeutic strategies currently adopted will be discussed, with a view to specific and targeted future treatments for patients with HCC.

## 2. Methods

An electronic search for literature was performed, updated on 15 November 2022, in PubMed/MEDLINE, Scopus, and Web of Science. We used a combination of the following keywords: (1) “liver steatosis” OR “NASH” AND “Insulin Resistance” OR “Metabolic Syndrome” OR “Diabetes” OR “Hepatocarcinoma” or (2) “NAFLD” OR “MAFLD” AND “Insulin Resistance” OR “Metabolic Syndrome” OR “Diabetes” OR “Hepatocarcinoma”.

Moreover, we ran a manual search for additional publications, which could have been missed through the electronic searches, on both clinical and animal studies.

We excluded all articles that were not available in English and where the full text was not accessible. A first screening was performed by reading titles and abstracts of the studies. Duplicate articles were removed.

## 3. NAFLD and NASH

Liver lipid storage is associated with visceral obesity, IR, and dyslipidemia in the complex picture of metabolic syndrome. NAFLD presents an accumulation of triglycerides (TG) within hepatocytes in the form of intracellular lipid droplets. TG are formed by esterification of free fatty acids (FA) and glycerol. Thus, the development of NAFLD is caused by an imbalance in hepatocellular FA metabolism [22,23]. The increased hepatic availability of FA is due to increased dietary intake, decreased inhibition of lipolysis at the level of adipose tissue caused by IR, as well as increased de novo hepatic lipogenesis [24]. Free fatty acids can undergo beta oxidation or TG esterification. Hepatic expression of CD36 fatty acid translocase is markedly increased in individuals with NAFLD. Furthermore, hepatic expression of adipocyte fatty acid-binding proteins (FABP), FABP-4 and FABP-5, is associated with an increase in intrahepatic TG [25,26]. An important accumulation of FA is due to de novo lipogenesis (DNL), a metabolic process that synthesizes new FA from excess glucose [27]. This pathway is an important contributor to hepatic lipid accumulation in the pathogenesis of NAFLD [28,29]. The activation of two transcription factors, sterol regulatory element binding protein-1c (SREBP-1c) and carbohydrate-responsive element binding protein (ChREBP), which are boosted by insulin and glucose responses to dietary carbohydrates [30], play a synergistically important role in the regulation of hepatic DNL. Additionally, in patients with NAFLD, a small amount of the FA pool is derived from dietary TG, which is associated with chylomicrons [31].

The most evocative theory in the pathogenesis of NAFLD is the “two-shot” hypothesis [32]. The first blow is insulin resistance caused by excessive FA flow in the liver. The second blow is inflammation associated with gut-derived endotoxin, oxidative stress, and mitochondrial dysfunction. It is closely related to the progression of NAFLD to NASH, which is characterized by excessive triglyceride accumulation (steatosis), inflammation, injury, and hepatocytes apoptosis that can lead to cirrhosis and HCC [33]. The remarkable prevalence of NAFLD and NASH could, in a short time, make both entities the most common predisposing factors of HCC in the coming years. The notion that HCC develops only in patients with cirrhotic NAFLD has long been challenged, as HCC has been increasingly recognized in non-cirrhotic patients with NASH [34]. Mohamad et al. observed in a small sample size that patients with NAFLD-HCC in the absence of cirrhosis had larger tumor diameters at diagnosis, higher rates of tumor recurrence, and worse survival outcomes than NAFLD-HCC patients with cirrhosis [35]. An Italian multicenter study enrolled 145 patients with NAFLD-related HCC between 2010 and 2012. An amount of about 50% of patients had no cirrhosis and all patients enrolled had NASH with advanced fibrosis rather than simple steatosis without fibrosis, suggesting that the stage of fibrosis might be relevant in the future risk of HCC in the absence of cirrhosis [36]. Furthermore, in a Japanese multicenter cohort of 87 patients with histologically proven NAFLD-related HCC diagnosed between 1993 and 2010, 72% of them had advanced fibrosis (F3 and F4), while 65% had at least moderate to severe necro-inflammatory activity [37].

## 4. NASH and HCC

NASH is characterized by metabolic dysregulation, chronic inflammation, and cell death in the liver, providing a favorable environment for the transformation of inflammation to cancer [38]. HCC is the second leading cause of years of life lost worldwide due to a cancer, which highlights the high burden of this disease [39]. NAFLD-related HCC is more common in older patients (with a mean age of 73 years), and is generally diagnosed at a later stage and associated with poorer survival than in viral hepatitis-related HCC [40]. NAFLD-related HCC develops in the absence of liver cirrhosis, differently from other liver diseases such as alcohol-related and autoimmune liver disease [41]. Currently, no screening protocols are available for HCC in patients with NAFLD without cirrhosis, which contributes to late diagnosis and management. The increase in free fatty acids in the cytoplasm of the hepatocyte is mainly induced by a high-fat diet, diabetes mellitus, IR, and reduced adiponectin in obese patients and induces oxidative stress within the cell due to an increase in reactive oxygen species (ROS) and cytokines production, such as TNF-alpha and IL-6 [42]. ROS production results in peroxidation of membrane lipids with mitochondrial and cellular damage. In addition, cytokine activation derived from the inflammatory trigger is also responsible for an increase in Hepcidin production [43]. High levels of Hepcidin reduce both intestinal iron absorption, due to the inhibition of Ferroportin (transmembrane iron transporter), and iron efflux from hepatocytes and macrophages. In NASH, intrahepatocyte accumulation of Fe2+ may contribute to inflammation and carcinogenesis [44] (Figure 1). Sorrentino et al. showed that iron intracellular deposition expressed as HIS (hepatocyte iron score) is much higher in NASH cases with HCC than in NASH patients without HCC [45,46]. Oxidative stress is also responsible for the activation of a kinase called JNK (c-jun amino acid kinase), a nuclear factor implicated in the reduction of cell apoptosis. Particularly, JNK is also activated indirectly by IR. Insulin resistance plays a crucial role at several levels in the cells, including insulin receptor desensitization, IRS protein and function suppression, PI3K cascades inhibition, and the inability to restrict transcriptional profiling of the Foxo1-activated gene, which may result from inhibition of IRS1 and IRS2. Under pathological conditions, the phosphorylation of IRS1 and IRS2 by p38α MAP kinase, JNK, mTOR, and protein kinase C (PKC) stimulate IRS protein degradation or inhibit IRS-associated PI3K pathway activation [47]. In fact, IRS1 and IRS2 represent the main endogenous mediators, in both the heart and the liver, in the activation of the PI3K → Akt signaling cascade [47,48]. IRS, in physiological conditions, activates the PI3K and Akt signaling pathway to maintain nutrient homeostasis and cardiac function, whereas many factors are involved in IR. During IR and diabetes development, due to IRS loss and the inactivation of the PI3K → Akt signaling pathway, the Foxo1 inhibitory mechanism is uncontrolled, due to Akt activation during feeding and/or insulin stimulation.

The Foxo1 dephosphorylation at conserved Akt phosphorylation sites (T24, S256, and S319) improves Foxo1 stability and transcriptional activity, increasing gluconeogenesis and inducing hyperglycemia. Foxo1-S 253 enhanced nuclear dephosphorylation was identified in T2DM animal models of both the liver and heart [49,50]. Deletion of Foxo1 in the liver of L-DKO mice and db/db mice led to a reduction of gluconeogenesis and diabetes amelioration, whilst Foxo1 deletion in the hearts of high-fat diet (HFD) mice prevented heart failure [51,52]. In addition, the overproduction of cytokines increases the production of nuclear factor NF-kB [49]. It seems that LDL oxidation promotes hepatic steatosis progression through several mechanisms, including the formation of oxidation-specific epitopes (OSEs). Enhanced oxidative stress causes lipid peroxidation, which can follow via enzymatic reactions (e.g., myeloperoxidase and 12/15-lipoxygenase) and non-enzymatic reactions (e.g., ROS) [53]. The peroxidation of membrane phospholipids results in their fragmentation and the production of degradation products, which may further modify both protein- and lipid-free amine groups, thus forming covalent adducts and creating OSEs, including malondialdehyde (MDA), 4-hydroxynonenal (4-HNE), phosphocholine on oxidized phospholipids (PC-OxPL), and oxidized cardiolipin (OxCL). These epitopes are transported by the oxidized low-density lipoprotein (OxLDL), modified proteins, microvesicles, and apoptotic cells, whose aspects seem to also be involved in NAFLD [53]. OSEs act as danger-associated molecular patterns (DAMPs) and are recognized by various pattern recognition receptors (PRRs), which represent a part of the cellular immune response to OSEs. The scavenger receptor (SR) family, such as CD36 Receptors, Toll-like receptors (TLRs), and the trigger receptor expressed on myeloid cells 2 (TREM2), are also known in the binding of certain lipid peroxidation adducts. In the liver, the presence of SRs and TLRs on Kupffer cells and their uptake of modified lipids have been described to induce liver injury, due to inflammation, during NAFLD progression. Moreover, specific TLR association to the progression of disease, such as TLR2, TLR4, TLR7, and TLR9, has been described [54]. For instance, TLR4 deletion seems to play a protective role against triglyceride liver accumulation in Ldlr −/−mice fed with an atherogenic diet, and in TLR4-deficient mice receiving a methionine–choline-deficient diet (MCD), NASH improvement was reported [55,56]. Moreover, TLR7 signaling activation has been described to prevent NAFLD progression, due to the reduction of hepatocytes’ lipid accumulation and autophagy. Intriguingly, 4-HNE and MDA potentially inhibit this process, thus exacerbating disease onset [57,58]. Though, as most studies have focused on bacterial ligand recognition, data on OSE-TLR signaling and fatty liver disease development are lacking [59]. Tregs, which are a subset of T cells essential for maintaining peripheral tolerance, preventing autoimmunity, and limiting chronic inflammatory disease have been shown to be more prone to mechanisms of programmed cell death, such as apoptosis, in a fatty liver than in a healthy liver. In addition, a recent study highlighted the importance of Tregs in NAFLD and NASH, providing clinical evidence of their role in the progression of liver damage. Several NADPH Oxidases (NOX) isoforms play an important role in supporting the progression of various chronic liver diseases [60], by increasing oxidative stress through the production of ROS, which is a major feature of liver damage [61]. It has been seen that metabolic syndrome can upregulate the expression of certain NOX isoforms, thereby inducing hepatic inflammation and oxidative stress [61]. In particular, it has been observed that a diet rich in fructose can upregulate the expression of both NOX2 and NOX4 in liver and adipose tissue. On the other hand, a diet rich in fat, in addition to causing hepatic lipid accumulation, upregulates NOX2, p47phox, and NOX4 in adipose tissue and NOX4 in the liver [62]. The gut microbiota is believed to be a driver of liver steatosis and inflammation [63]. Gut flora dysbiosis is involved in the development of several liver diseases, including simple steatosis, NASH, and NAFLD-HCC. These patients have a significantly lower abundance of Clostridium than healthy individuals [64]. The percentage of alcohol-producing bacteria in the gut is also increased in patients with NASH, leading to increased blood ethanol concentrations. Finally, microbial metabolites, in addition to the gut microbes themselves, represent another interesting link between the microbiome and NAFLD [65]. Several metabolites, including amino acids, short chain-fatty acids (SCFAs), and bile acids, in the gut, circulation, and liver tissues, have been identified to ameliorate NAFLD and NAFLD-HCC. Bile acids are closely associated with NASH and NASH-HCC. Bile acids are other important metabolites which link the gut microbiota to liver disease. Furthermore, bile acids can alter their receptor, the farnesoid X receptor (FXR), to modulate the development of NASH [59]. In fact, FXR activation is known to reduce triglyceride levels and inhibit fatty acid synthesis and uptake in the liver [66]. In addition, a role of FXR in reducing inflammation has emerged [67]. Obeticholic acid, an activator of FXR, is known to significantly improve fibrosis and disease severity in patients with NASH [68]. In general, primary bile acids secreted by the liver are not bound to intestinal microorganisms. Therefore, these unbound bile acids are reabsorbed to form secondary bile acids and are returned to the liver for detoxification [69]. Dysregulated crosstalk between bile acid and the microbiome can impair this process, thus contributing to inflammation and the development of HCC. Secondary bile acids have also been shown to regulate immune function and HCC development. For example, deoxycholic acid (DCA) induces NASH-related HCC by activating mTOR [70]. Increased DCA in the liver can also trigger the Senescence-Associated Secretory Phenotype (SASP) in HSCs. This, in turn, leads to the production of proinflammatory cytokines and cancerogenic factors in the liver, thus promoting HCC [71]. Antibiotic treatment reduces the production of DCA-producing bacteria, which in turn reduces the development of HCC [72]. This suggests that the DCA-SASP axis in HSCs promotes the development of obesity-associated HCC. Importantly, blocking DCA production or modifying gut microbiota may reduce the development of HCC.

## 5. NAFLD and Diabetes: A Single Combination Therapy?

Weight loss and lifestyle changes are the main approaches to improve symptoms/signs of NAFLD, but antihyperglycemic drugs can be used to address conditions associated with NAFLD such as dyslipidemia, IR, liver apoptosis, inflammation, and fibrosis [73]. NAFLD frequently occurs in patients with diabetes, obesity, and/or metabolic syndrome and enhances cardiovascular risks, so treating these conditions is as important as NAFLD treatment in itself [74].

Statins are the first medications in treating patients with NAFLD, even once the disease has progressed to NASH, as they reduce cardiovascular risk and liver enzymes in patients with elevated alanine aminotransferase at the baseline [61]. In addition, non-statin hypolipidemic therapies such as ezetimibe, bile acid sequestrants, PCSK9 inhibitors, and omega-3 fatty acids can reduce residual lipid risks and may confer liver benefits [75]. Hypolipidemic therapy is essential in patients with NAFLD and NASH, who are often at high risk for cardiovascular disease, have already had a cardiovascular event, or are at high risk for metabolic syndrome and diabetes. In recent years, several antihyperglycemic drugs have been studied in patients with NAFLD/NASH, such as pioglitazone, sitagliptin, GLP-1 receptor agonists, and SGLT2 inhibitors. It has been reported that pioglitazone and high-dose vitamin E might improve the histology of patients with NASH [74]. Nevertheless, metformin does not recover the liver histology of patients with NAFLD [76], and ursodesoxycholic acid (UDCA) does not improve liver histology, inflammation, or fibrosis in patients with NASH [71]. Vitamin E is well known as a free radical scavenger and has been predicted for the treatment of NASH. In the PIVENS (Pioglitazone versus Vitamin E versus Placebo for the Treatment of Nondiabetic Patients with Nonalcoholic Steatohepatitis) study, vitamin E (800 mg/day) was superior to a placebo in NASH histology improvement in adults without diabetes and cirrhosis [77,78]. According to a random-effects model analysis of the five studies, vitamin E significantly reduced serum hepatobiliary enzymes, hepatic steatosis, inflammation, and hepatocellular swelling compared with the control group [79]. Based on the PIVENS study, vitamin E is currently recommended only for patients with biopsy-proven NASH without diabetes and is associated with histologic improvement regardless of diabetic status [80]. However, long-term or high-dose Vitamin E intake is potentially dangerous. Vitamin E treatment may increase mortality from all causes, prostate cancer, and hemorrhagic stroke [81,82]. Vitamin E for the treatment of NASH should be considered effective with a lower dose (300–400 mg/day instead of 800 mg) [83]. Pioglitazone (PGZ), is an antidiabetic agent widely used for the treatment of T2DM and acts as insulin sensitizers, helping to regulate blood glucose and IR [84]. The class drug of thiazolidinedione (TZD) does not cause hypoglycemia in single therapy and does not present a contraindication in patients with chronic kidney disease [85,86]. PGZ modulates metabolic pathways through peroxisome proliferator-activated receptor nuclear transcription factor gamma (PPARγ) binding and target gene expression regulation [85]. In patients with NASH and T2DM, PGZ reduces hepatic steatosis, inflammation, and serum levels of alanine aminotransferase (ALT) and aspartate aminotransferase (AST) and improves liver fibrosis [86]. In mouse models, PGZ reduces hepatic gluconeogenesis, improves insulin sensitivity in the liver and other peripheral tissues, and also ameliorates hepatic fibrosis [80]. Obeticolic acid (OCA) is a selective agonist of the farnesoid X receptor (FXR), a nuclear receptor that can regulate hepatic glucose and lipid metabolism, inflammation and lipoprotein composition, and bile acid synthesis [87]. OCA increases insulin sensitivity and reduces the markers related to liver inflammation and fibrosis in patients with NAFLD [88]. Its effect also results in weight loss in patients with NASH, thus exerting additional beneficial effects on serum ALT and liver histology [89]. It also significantly improves fibrosis in patients with NASH [90]. Bempedoic acid (ETC-1002), an ATP citrate lyase (ACLY) inhibitor that lowers LDL cholesterol, was recently approved by the US FDA for the treatment of heterozygous familial hypercholesterolemia (HeFH) and established atherosclerotic cardiovascular disease (ASCVD). A recent study evaluated the effect of the ACLY inhibitor bempedoic acid in the NASH animal model induced by a long-term high-fat diet [91]. This study identified a promising role for bempedoic acid in improving metabolic disorders and NASH. Treatment with ETC-1002 alleviated long-term HFD-induced NASH through the inhibition of body weight gain, improvement of glycemic control, reduction of hepatic triglycerides and total cholesterol, and modulation of inflammatory and fibrotic genes. Further investigations are needed to investigate its potential role for the treatment of NASH. Glucagon-like peptide-1 receptor (GLP-1R) agonists are effective antihyperglycemic drugs both in animal models and in patients with T2DM [92,93]. Among the GLP-1Rs, liraglutide received FDA approval in 2020 for obese patients’ treatment, based on its lasting benefits in weight loss [94]. In patients with NAFLD and NASH, it reduces liver fat content, improves histological resolution and decreases serum levels of liver enzymes without worsening fibrosis [95]. Liraglutide effects on weight loss and on cardiovascular risk reduction are essential for the treatment of NAFLD, due to its lipotoxicity and IR positive effects [96]. Studies have shown that liraglutide protects pancreatic beta cells from apoptosis through AKT-mediated survival signaling [90], improves insulin sensitivity by activating AMP-activated protein kinase (AMPK), and by reducing hepatic steatosis through lipid transport, β-oxidation, and autophagy regulation [91]. Some pilot studies involving SGLT2i in patients with NAFLD are being conducted in Western countries (NCT02696941) or Asia (NCT02875821, NCT02964715). The effect of SGLT2 inhibitors was compared with other anti-diabetic drugs (e.g., metformin, sulphonylureas) (NCT02696941, NCT02649465) [97]. The effects of empagliflozin treatment on hepatocellular lipid content, hepatic energy metabolism, and body composition are now being studied in a multicenter, RDBPCT, interventional, exploratory pilot study in patients with newly diagnosed T2DM (NCT0263797). Selonseritib, a first-in-class inhibitor of signal-regulating kinase 1, has been proposed as a treatment for fibrotic diseases [98]. Selonseritib inhibits phosphorylation and activation of ASK 1 by binding to the catalytic kinase domain of ASK1 in mouse models. ASK1, a serine/threonine signaling kinase, causes phosphorylation of p38 mitogen-activated kinase and c-Jun N-terminal kinase (JNK) leading to the activation of stress response pathways that exacerbate hepatic inflammation, apoptosis, and fibrosis [99]. In murine models of NASH, it significantly improves not only hepatic steatosis and fibrosis associated with NASH, but also cholesterol and bile acid levels, and lipid metabolism [100]. In a phase 2 clinical trial involving patients with NASH and stage 2–3 of liver fibrosis, it has been described to prevent inflammation, fibrosis, excessive apoptosis, and progression to cirrhosis [96]. Currently, phase 3 clinical trials on patients with NASH and advanced fibrosis have confirmed liver histology amelioration, though not affecting fibrosis [101,102]. Simtuzumab (SIM) is a monoclonal antibody targeting the lysyl oxidase-like enzyme 2 (LOXL2), which catalyzes the cross-linking of collagen and elastin, leading to a remodeling of the extracellular matrix [103]. SIM inhibits the synthesis of growth factors including connective tissue growth factor (CTGF/CCN2) and TGFβ1, and reduces liver fibrosis. However, in phase 2b clinical trials in subjects with NASH-induced advanced fibrosis, no effect on the improvement of fibrosis and cirrhosis confirmed by liver collagen content was found [104]. Cenicriviroc (CVC) is a potent CCR2 and CCR5 antagonist currently in clinical development for the treatment of liver fibrosis in patients with NASH [103]. CVC reduces levels of inflammation markers including IL-1β and IL-6 and exerts antifibrotic activities [101]. It received Fast Track designation from the FDA in 2015 as a promising therapy for NASH and liver fibrosis. In the phase 2b study in subjects with NASH and stage 2–3 fibrosis, CVC showed improvement in liver fibrosis without worsening NASH [105]. Phase 3 clinical trials are currently underway to evaluate and confirm the efficacy and safety of CVC for the treatment of liver fibrosis in patients with NASH [105]. It is acceptable to consider that NAFLD is caused by a concert of various factors including nutritional factors, gut microbiota, and genetic and epigenetic factors as well as adipokines and hepatokines. To find an appropriate treatment, it is necessary to look at various factors in a broader perspective by making use of the classification according to NAFLD etiology [105,106]. Gut microbiota is influenced by an incorrect lifestyle, from both a qualitative and quantitative point of view, as well as by a direct (through chemical mediators produced by bacteria) and an indirect (through interference with biochemical metabolic pathways) contribution, and is further involved in T2DM development and NAFD progression [107]. However, this association is still not clear. By contrast, it has been suggested that the microbiome, by inducing systemic inflammation, insulin resistance, and oxidative stress enhancement, might represent a key factor in metabolic dysfunction and T2DM development [108,109]. In fact, microbial dysbiosis, with a consequential increased ethanol production, is responsible for liver toxicity and increased intestinal permeability, secondary to tight junction loss. Thus, gut-derived pathogen-associated molecular patterns (PAMPs) (e.g., lipopolysaccharide) are driven by blood flow into the liver. As a consequence, there is an increase in liver inflammation and fibrosis, due to the activation of proinflammatory pathways. In addition, gut microbiota, associated with increased choline metabolism, lead to an increase in liver triglycerides storage, due to the lack of VLDL excretion. Moreover, dysbiosis has been related with decreased secretion of fasting-induced adipocyte factors [FIAF and angiopoietin-like protein 4 (ANGPTL4)], which are in turn involved in endothelial lipoprotein lipase inhibition and therefore, the lack of liver triglycerides hydrolyzation from VLDL particles [110]. Finally, the increased number of substrates, in particular short-chain fatty acids, upregulate glucogenesis and lipogenesis, with a consequent hepatic FFAs accumulation promotion through AMPK inhibition [111].

Lifestyle modification represents the first line of treatment in T2DM patients. In fact, patients at an increased risk of developing T2DM live a sedentary life and present an unbalanced diet, with an increased intake of simple carbohydrates and saturated fats. As suggested by the majority of scientists, Mediterranean, but also vegan and vegetarian dietary patterns consumption should be implemented in public health strategies to improve glycemic control in T2DM individuals [112]. In addition, it is crucial to add aerobic or resistance exercises training along with diet, as this association has proven to prevent cardiovascular and dysmetabolic diseases. In fact, low-moderate physical activity has been reported to increase treatment effectiveness in obtaining better glycemic control [113].

## 6. Diabetes and HCC

HCC is a neoplasm with a high mortality incidence and is affected by viral and non-viral factors which tend to worsen the patient’s prognosis regardless of liver function [114,115,116,117,118]. Moreover, HCC remains the leading cause of cancer-related death among patients with T2DM. The risk of HCC recurrence is 2.5–4-fold higher in patients with T2DM, independently of the presence of cirrhosis or of the etiology of the underlying liver disease [119,120,121,122,123]. According to Torres et al. [124], the pathophysiology of the association between T2DM and HCC is not completely understood. Consequently, it is an arduous challenge to identify genes and pathways involved in the relationship between T2DM and HCC to clarify their functions, prognostic roles, and therapeutic perspectives. Some key genes might play a critical role in both T2DM and HCC. Particularly, 10 hub genes—CCNA2, CCNB1, MAD2L1, BU1B, RACGAP1, CHEK1, BUB1, ASPM, NCAPG, TTK—have a strong association with lower overall survival in liver cancer patients. Additionally, four of the aformentioned genes—CCNA2, CCNB1, CHEK1, BUB1—have reduced expression in metformin-treated samples [125,126]. Obviously, factors such as Kruppel-like factor 6, abnormal methylation, and immune dysregulation might explain the dysregulation of hub genes. An acquired CD44 phenotype in macrophages, for example, is associated with T2DM-HCC and lobular inflammation [127]. A very important role is also played by miRna and lncRna. More properly, lncRNA LINCO1572 is aberrantly upregulated in HCC tissues, especially in those with concurrent T2DM, and is associated with advanced tumor stage, increased blood HbA1c level, and shortened survival time [121]. Regarding miRNAs, the expression of miR34 induces hepatocyte senescence, instead of miR15a and miR16-1, which can prevent HCC in both AKT/RAS and c-Myc pathways [121]. MiR-122 is suppressed in HCC cells and its overexpression can increase HCC cells’ radiosensitivity [123]. Also, Metabolomics have emerged as a powerful tool for the discovery of novel biomarkers for early detection of HCC in T2DM patients [124]. Targeted metabolite analyses confirmed that serum benzoic acid and citrulline are increased and creatine is decreased in patients with T2DM and HCC. The combination of these serum metabolites and alpha fetoprotein (AFP) might be useful in the surveillance of HCC in T2DM patients [124]. Concerning pathophysiologic mechanisms, it seems that IR and activation of the insulin receptor and insulin-like growth factor 1 (IGF-1) signaling pathways are the main determinants in the initiation and progression of hepatocarcinogenesis. In fact, it has been shown that hepatoma cells overexpress IGF-1 and insulin receptor substrate-1, suggesting their importance in HCC development [125]. IR causes inflammation, oxidative stress, and stimulation of cellular pathways that stimulates cellular growth and cellular proliferation. In addition, it leads to a systemic redistribution of the substrate that improves tumor growth.

On the other hand, unbalanced IGF/IGF-1R signaling can promote cancer cell proliferation and activate cancer reprogramming in tumor tissues, especially in the liver [126,127], by the activation of multiple cytokine pathways. The latter include phosphoinositide-3-kinase/AKT/mTOR and mitogen activated protein kinases, which modify the cell cycle and thus, cellular proliferation. Additionally, IRS-1 may play a role in preventing TGF-β–mediated apoptosis. The complex integration of carcinogenic mechanisms in NAFLD such as chronic inflammation, lipotoxicity, and high insulin levels can lead to a higher HCC risk, especially among patients with underlying T2DM. A recent retrospective analysis based on a large ethnically diverse cohort of patients with NAFLD reported that HCC risk was more than 7-fold higher in patients with NAFLD compared with matched controls (adjusted HR [aHR]: 7.62; 95% CI: 5.76–10.09), whereas subjects with diabetes had three times higher the risk compared with nondiabetics (aHR: 3.03; 95% CI: 2.52–3.64) [128]. In the study by Doycheva et al. [128], patients with diabetes and NASH had the highest risk of developing HCC (odds ratio [OR] 1.68; 95% confidence interval [CI] 1.52–1.86), evaluated with non-diabetics.

Diabetes associated with NASH, cryptogenic cirrhosis, hepatitis C, and alcoholic liver disease improved the risk of HCC incidence, compared to non-diabetic individuals. Though, in patients with chronic hepatitis B or primary biliary cholangitis, diabetes did not increase HCC risk. Additionally, the risk of HCC appears to be higher in patients with long standing and poorly controlled diabetes. Patients with good glycemic control (defined as HbA1c < 7% for >80% time) were associated with a 32% decreased risk of HCC than in patients who had suboptimal glycemic control (HR, 0.68; 95% CI, 0.60–0.77; *p* < 0.0001). Patients with diabetes complications were associated with a 24% increased risk of HCC than in patients without diabetes complications (HR, 1.24; 95% CI, 1.12–1.38; *p* < 0.0001) [125,126,127].

Patients with NAFLD at high risk of HCC should be identified because of the reversibility of NAFLD related to weight loss. For this reason, screening for liver disease in T2DM is recommended according to European guidelines [129]. American guidelines, while not advocating a screening protocol, suggest using non-invasive markers of fibrosis for risk stratification, such as the Fibrosis-4, NAFLD fibrosis score, AST/platelet ratio index, enhanced liver fibrosis (FIB-4, NFS, APRI, ELF) [130]. To test the ability of individual fibrosis scores and the European screening algorithm to predict 11-year incident cirrhosis/HCC in an asymptomatic community cohort of older people with T2DM, the Edinburgh Type 2 Diabetes Study [129] investigated men and women with T2DM (*n* = 1066, aged 60–75 at baseline). Forty-three out of 1066 participants with no baseline cirrhosis/HCC developed incident disease. All scores were significantly associated with incident liver disease by an odds ratio (*p* < 0.05). The ability of the risk-stratification tools to accurately identify those who developed incident cirrhosis/HCC was low with low-positive predictive values (5–46%) and high false-negative and positive rates (up to 60% and 77%), respectively. When fibrosis risk scores were used in conjunction with the European algorithm, they performed modestly better than when applied alone [129]. Prevention and treatment options for HCC include lifestyle change, dietary supplement, modulation of gut microbiota, anti-inflammation and anti-oxidative stress medicines, and anti-obesity and anti-diabetic treatments. Adherence to a healthy diet with a higher intake of cereal fiber, polyunsaturated fat, and nuts associated to physical activity has been correlated to a reduced risk of HCC in T2DM patients through the suppression of hepatic lipogenesis and IR and the decrease of plasma levels of PCKS9 and inflammatory molecules [130]. Given the increased risk of HCC in patients with T2DM, now it is interesting to also see if anti-diabetic medications could have a role in the incidence and outcome of HCC. According to some authors, metformin has a protective role in HCC development. Zhang et al. [131], combining three cohort studies and four case-control studies with more than 16,000 diabetic subjects, showed that metformin treatment was associated with a 76% reduced risk of HCC. This effect seems to be related to the activation of AMPK and the inhibition of m-TOR pathways (relative risk [RR] 0.24, 95% CI 0.13–0.46, *p* < 0.001) [132,133,134,135,136,137,138]. Unlike metformin and TZDs, sulfonylureas have consistently been reported to significantly increase the risk of HCC, especially for patients treated with second generation sulfonylureas [139]. Insulin is a potent mitogen and in the past decade, increasing evidence has described a higher risk of HCC incidence in diabetic patients treated with insulin [140,141]. Increased risks of HCC were associated with the use of insulin (OR 3.73, 95% CI 2.52–5.51), with a higher risk associated with longer treatment duration (OR 2.52 for <1 year, 5.41 for 1–2 years, and 6.01 for ≥2 years; *p* < 0.001), and sulfonylureas use (OR 1.39, 95% CI 0.98–1.99) [142]. Similarly, a recent study showed a 2-fold increased risk of HCC for patients treated with insulin (OR 1.9, 95% CI: 0.8–4.6). These results are in accordance with those from the metanalysis by Singh et al., who found an increased risk of HCC for insulin users (OR 2.61, 95% CI 1.46–4.65) [143]. Data on the more novel glucose-lowering agents (i.e., DDP-4 inhibitors, SGLT2-inhibitors, and GLP1-RAs) are preliminary and lack large clinical studies.

However, Chung et al. [131] proposed, for patients with T2DM and chronic liver disease, a treatment algorithm centered on GLP-1 receptor agonists, DPP-4 inhibitors, and SGLT-2 inhibitors. These class drugs should be preferred even in patients with mild-to-moderate hepatic impairment due to their low risk of hypoglycemia and the potential protective effect on HCC development in vitro [144,145,146,147]. In conclusion, anti-angiogenetic drugs such as sorafenib, regorafenib, and lenvatinib have a therapeutic effect on HCC, even if some studies showed that T2DM patients with HCC who received metformin are resistant to sorafenib [148].

## 7. HCC and Drug Treatment Update

HCC treatment remains an important problem in the approach and management of the patient, due to the lack of surgical and pharmacological possibilities [149,150]. In recent years, the pharmacological therapy of HCC is on the rise due to the introduction of new molecules, particularly immune checkpoint inhibitors (ICIs) [151].

Sorafenib, a multikinase inhibitor approved in 2007, which is still indicated as a first line in CHILD A cirrhosis patients with inoperable HCC [152,153], has been joined by the introduction of new drugs approved by the international guidelines [154,155].

In 2018, lenvatinib, a vascular endothelial growth factor receptor (VEGFR), platelet-derived growth factor receptor (PDGFR), KIT, RET multikinase, and fibroblast growth factor receptor (FGFR) activities inhibitor, was also approved with the same clinical indications [156,157]. The comparison trial revealed non-inferiority median overall survival (OS) between the lenvatinib and sorafenib group. Moreover, lenvatinib was associated with a higher objective response rate (ORR) of 24.1%, a better progress free survival (PFS) of 7.4 months, and a longer median time to progression (TTP) of 8.9 months [158].

The first combination therapy, approved as first line therapy in CHILD A cirrhosis patients, was atezolizumab plus bevacizumab, a target programmed cell death-1(PD-L1) blocker and VEGF inhibitor, respectively. The combination therapy showed significantly better median PFS of 6.8 months vs. 4.3 months for sorafenib, a better median OS and ORR of 19.2 months (versus 13.4 months for sorafenib), and 30%, compared with sorafenib (11%), respectively [159,160]. The drug combination did not show a significant increase in adverse events, but a slightly increased rate of hypertension, proteinuria, and increased AST and ALT serum levels were detected. However, bevacizumab is associated with a higher bleeding risk. A recent phase III trial (HIMALAYA) showed promising results from the association between tremelimumab, a cytotoxic T-lymphocyte-associated protein 4 (CTLA-4 inhibitor), and durvalumab (anti-PD-L1). This study showed the better OS and ORR for the drugs combination rather than durvalumab or sorafenib alone with a favorable safety [161]. Their use could be a first-line option in case of intolerance to atezolizumab plus bevacizumab [162].

As second line therapies, several molecules are available in single or in combination. Regorafenib and cabozantinib are two multikinase inhibitors. However, similar to Sorafenib, Regorafenib showed more effects on angiogenesis and tumor growth with a better PFS, OS, ORR, and disease control rate (DCR) than the placebo [163]. Similar results have been achieved by Cabozantinib, whose efficacy was associated to the early reduction of alpha fetoprotein [164,165].

Conflicting results emerged by trials conducted on Ramucirumab (a recombinant immunoglobulin G subclass 1) and Pembrolizumab (anti PD-1). However, the latest evidence enabled the inclusion of both in the second line treatment [166,167,168,169].

An interesting antibodies combination therapy, Nivolumab plus Ipilimumab (PD-1 and CTLA-4 inhibitor), was evaluated in a phase I/II CHECKMATE-040 trial [170]. In 148 sorefanib treated CHILD A cirrhosis patients, the best arm of the trial showed 8% and 24% of complete and partial responses, respectively. Currently, this regimen is investigated as a first-line therapy in the phase III trial [171].

Several trials are ongoing to evaluate the efficacy of new molecules. A phase II–III Chinese trial enrolled 667 patients who were randomized 1:1 to receive donafenib (a novel multikinase inhibitor) or sorafenib. The results showed better OS and safety of donafenib that could be proposed as an alternative first line monotherapy [172].

Globally, the international guidelines recommend, as the first line, the association of atezolizumab–bevacizumab or tremelimumab plus durvalumab, depending on drug tolerance. In the second line, monotherapy with lenvatinib, sorafenib, cabozantinib, regorafenib, or ramcirumab is recommended. For patients who had received monotherapy in the first line (sorafenib, lenvatinib), the combination nivolumab plus ipilimumab or alternatively, a single molecule (pembrolizumab or regoarafenib, cabozantinib, ramcirumab) is preferable [173]. A recent metanalysis confirmed that combined targeted drug and immunotherapy significantly improved survival compared with targeted monotherapy, although with a higher rate of adverse events [174].

## 8. Conclusions

In the last two decades, the proportion of HCC patients with non-viral etiology has increased rapidly; in fact, the importance of HCC derived from NAFLD/NASH is emerging. The comorbidities of NAFLD as obesity, type 2 diabetes, pre-diabetes, and cardiovascular disease are risk factors that contribute to the onset and progression of HCC [175]. Pathogenic factors such as abnormal metabolites, inflammatory factors, and immune modulations are the mechanisms underlying the metabolic dysfunction associated with the pathogenesis of HCC. Therapy of patients with NAFLD/NASH is generally performed using hypolipidemic and hypoglycemic drugs. The side effects that appear with long-term use cannot be ignored. Therefore, appropriate therapeutic targets and FDA-approved therapies are urgently needed. In addition, there are several therapeutic options that show promising effects on HCC. However, the benefits of these treatments are still limited. Further, clinical trials are awaited to explore potential treatments for HCC associated with metabolic diseases.

## Figures and Tables

**Figure 1 biomedicines-11-00468-f001:**
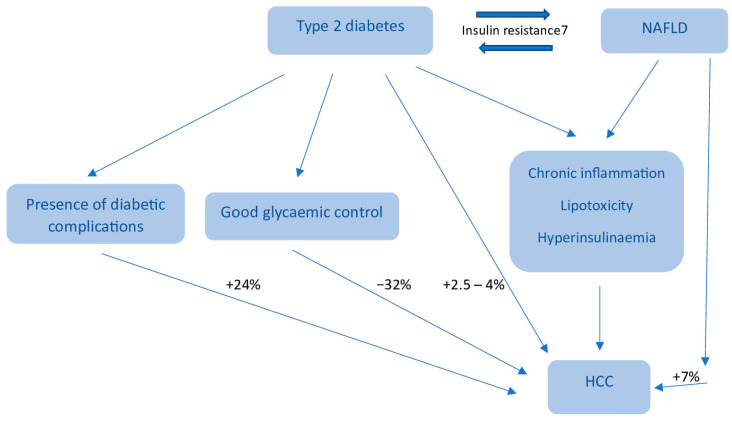
HCC development risk in type 2 diabetes and NAFLD.

## Data Availability

Not applicable.

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
