# Peer review of "Non-alcoholic Fatty Liver Disease (NAFLD), Type 2 Diabetes, and Non-viral Hepatocarcinoma: Pathophysiological Mechanisms and New Therapeutic Strategies"

_biomedicines, 2023, doi:10.3390/biomedicines11020468_

Round 1

Reviewer 1 Report

It does not typically follow the guideline:

Introduction

Materials and methods

Results 

Discussion 

Author Response

It does not typically follow the guideline:

Introduction

Materials and methods

Results 

Discussion 

R. We thank the reviewer for his/her suggestions. The organization of the paragraphs of a review differs from an original article due to the nature of the paper which describes and comments on the resultsof the literature. For this reason we have structured the review with the introduction and the methods, in which we have illustrated the method of collecting the studies. Subsequentaly we have preferred to list the paragraphs according to the different sections concerning the topic of the review. We have not indicated a result paragraph, as we usually refer this concept to experiments illustrated in the original articles. We have also made comments in the context of these paragraphs with the aim of including a discussion of each specific topic. Therefore, we ended the paper with a short paragraph of conclusions.

Reviewer 2 Report

The authors wrote comprehensive review describing the problems of pathogenesis and of current therapeutic approaches of NASH related disorders with the main focus on nonviral hepatocellular carcinoma. But there are some minor points the I should address to:

  1. Not all the abbreviations are deciphered: OCE (line 151), NOX (line 162), SASP (line 193), THCC (line 423) etc. Check and revise.
  2. Line 144. Is there any prove that hyperinsulinemia in the insulin resistance state indeed activates the consensus downstream insulin signaling kinase, such as protein kinase Akt? In normal state yes, it does, but in insulin resistance?
  3. Please, list some of the examples of OSE epitopes (line 150), and is there any data that they do active Tregs?

Author Response

The authors wrote comprehensive review describing the problems of pathogenesis and of current therapeutic approaches of NASH related disorders with the main focus on nonviral hepatocellular carcinoma. But there are some minor points the I should address to:

  • Not all the abbreviations are deciphered: OCE (line 151), NOX (line 162), SASP (line 193), THCC (line 423) etc. Check and revise.

Author: We wish to thank the reviewer for his/her comment. Accordingly, we modified the text by specifying the abbreviations.

  • Line 144. Is there any prove that hyperinsulinemia in the insulin resistance state indeed activates the consensus downstream insulin signaling kinase, such as protein kinase Akt? In normal state yes, it does, but in insulin resistance?

Author: Accordingly, we modified the text.

Insulin resistance occurs at multiple levels in cells, from the cell surface to the nucleus, including desensitization of the insulin receptor, suppression of IRS protein and function, inhibition of PI3K cascades, and inability to restrict transcriptional profiling of the Foxo1-activated gene, which may result from inhibition of IRS1 and IRS2. IRS1 and IRS2 each contain 40 potential serine/threonine sites, which are phosphorylated by p38α MAP kinase, JNK, mTOR, and protein kinase C (PKC), stimulating IRS protein degradation or inhibiting IRS-associated PI3K activation under pathological conditions. IRS1 and IRS2 are the main endogenous mediators that activate the PI3K→Akt signaling cascade in the liver and heart of animals. Normal expression and function of IRS activating the PI3K and Akt signaling pathway is essential for animals to maintain nutrient homeostasis and cardiac function, whereas many factors can cause insulin resistance. During the development of insulin resistance and diabetes mellitus, following the loss of IRS and the inactivation of the PI3K → Akt signaling pathway, the inhibitory mechanism of Foxo1, by activation of Akt during feeding or insulin stimulation, is uncontrolled. Therefore, defosphorylation of Foxo1 at conserved Akt phosphorylation sites (T 24, S 256 and S 319) enhances Foxo1 stability and transcriptional activity, stimulating gluconeogenesis and causing hyperglycemia. An increase in nuclear defosphorylated Foxo1-S 253 was detected in the liver and heart of animals with type 2 diabetes. Deletion of Foxo1 in the liver of L-DKO mice and db/db mice, reduction of hepatic glucose production and improvement of diabetes and Foxo1 deletion in the hearts of HFD mice prevented heart failure.

  • Please, list some of the examples of OSE epitopes (line 150), and is there any data that they do active Tregs?

Author:  Accordingly, we modified the text.

There appears to be LDL oxidation contributing to the progression of hepatic steatosis by multiple mechanisms, including the formation of OSE. Increased oxidative stress results in lipid peroxidation, which can occur via enzymatic reactions, such as myeloperoxidase and 12/15-lipoxygenase, and nonenzymatic reactions, such as reactive oxygen species (ROS). Lipid peroxidation of membrane phospholipids results in their fragmentation and the generation of degradation products that can further modify the free amine groups of proteins and lipids, forming covalent adducts and creating oxidation-specific epitopes (OSE), including malondialdehyde (MDA), 4-hydroxynonenal (4-HNE), phosphocholine on oxidized phospholipids (PC-OxPL), and oxidized cardiolipin (OxCL). These epitopes are carried by oxidized low-density lipoprotein (OxLDL), modified proteins, microvesicles, and apoptotic cells, aspects that have been shown to be present during NAFLD. Oxidation-specific epitopes (OSEs) act as danger-associated molecular patterns (DAMPs) that are recognized by several pattern recognition receptors (PRRs) as part of the cellular immune response toward OSEs. Receptors known to bind to certain lipid peroxidation adducts are the scavenger receptor (SR) family such as CD36, Toll-like receptors (TLRs) and the trigger receptor expressed on myeloid cells 2 (TREM2). In the liver, the presence of SRs and TLRs on Kupffer cells and their uptake of modified lipids have been shown to cause inflammation, thus leading to liver damage during the progression of NAFLD. In relation to NAFLD, the involvement of certain TLRs in disease progression, such as TLR2, TLR4, TLR7, and TLR9, has been described. For example, the absence of TLR4 in Ldlr -/- mice on an atherogenic diet has been shown to protect against triglyceride accumulation in the liver and features of NASH were improved in TLR4-deficient mice receiving a methionine-choline-deficient diet (MCD). In addition, activation of TLR7 signaling has been found to reduce lipid accumulation and autophagy in hepatocytes, thereby preventing the progression of NAFLD. Interestingly, 4-HNE and MDA potentially inhibit this process, thereby exacerbating the onset of the disease. However, a direct link between OSE-TLR signaling and the development of fatty liver disease is lacking because most studies focus on bacterial ligand recognition.

Reviewer 3 Report

The manuscript submitted by Veteran et al., is aiming to discuss the axis of NAFLD-T2DM-HCC including the mechanistic aspects informing this axial relationship. 

This is an interesting topic and the manuscript is providing an extensive and logically arranged review of evidence regarding the topic. 

The reviewer would like to note a few points for the improvement of the manuscript:

1. What was the rationale/approach in terms of the selection of articles that were considered towards the points addressed by the manuscript?

2. The authors could consider a PRISMA figure to indicate the strategy by which they selected the articles they used. 

3. One important issue in the management and progression of T2DM is the microbiome. While this is a complex relationship it would be important and most useful while supporting a more comprehensive approach if this topic is briefly discussed. A useful paper towards this end is the following: 

Sikalidis, A.K.; Maykish, A. The Gut Microbiome and Type 2 Diabetes Mellitus: Discussing A Complex Relationship. Biomedicines 2020, 8, 8. https://doi.org/10.3390/biomedicines8010008.

4. When discussing the therapeutic strategies especially in the context of T2DM it is important to address the dietary components (nutrient categories) of the meals and the dietary patterns overall. There is obviously extensive literature on the topic but it would be more appropriate and comprehensive not to limit the approach to the pharmacological one only.  

5. A visual summarizing the major points of the narrative would be very powerful and useful. 

6. Proofreading would be suggested. 

Author Response

The manuscript submitted by Veteran et al., is aiming to discuss the axis of NAFLD-T2DM-HCC including the mechanistic aspects informing this axial relationship. This is an interesting topic and the manuscript is providing an extensive and logically arranged review of evidence regarding the topic. The reviewer would like to note a few points for the improvement of the manuscript:

  1. What was the rationale/approach in terms of the selection of articles that were considered towards the points addressed by the manuscript?

- Authors:

We ran an electronic search in PubMed/MEDLINE, Scopus, and Web of Science for literature updated to 15/11/2022. A combination of the following keywords was used: (1) “liver steatosis” OR “NASH” AND “Insulin Resistance” OR “Metabolic Syndrome” OR “Diabetes” OR “Hepatocarcinoma” or (2) “NAFLD” OR “MAFLD” AND “Insulin Resistance” OR “Metabolic Syndrome” OR “Diabetes” OR “Hepatocarcinoma”. Moreover, we conducted a manual search of references to relevant studies and animal studies, to find additional publications that might have missed through electronic searches only. Articles for which the full text was not accessible or not available in English were excluded. The potential study found were than matched between the two searches. Duplicate articles were removed, and a first screening was performed by reading only the titles and abstracts of the studies.

  1. One important issue in the management and progression of T2DM is the microbiome. While this is a complex relationship it would be important and most useful while supporting a more comprehensive approach if this topic is briefly discussed. A useful paper towards this end is the following:

- Authors: Accordingly, we modified the text, by adding the suggested citation.

Dysbiosis Lifestyle plays an important role in the qualitative and quantitative change of the intestinal microbiota, which contributes, both, directly (through chemical mediators produced by bacteria) and indirectly (through interference with the biochemical metabolic pathways), to the development of T2DM and NAFLD. The association between Gut Microbiome and the development of T2DM is not clear not completely understood. However, some authors outline that microbiome, through an enhancement of systemic inflammation, insulin resistance and oxidative stress, could drive to a metabolic deregulation, which in turns are key factors in the development of T2DM. In fact, the increase in intestinal ethanol production, due to microbial dysbiosis, leads to both liver toxicity and an increase in intestinal permeability, secondary to the loss of the tight junction. Thus, gut-derived pathogen-associated molecular patterns (PAMPs), such as lipopolysaccharide, reach the liver through blood flow. As a result, proinflammatory pathways activate specific binding tool such as receptors and lead to liver inflammation and fibrosis. Furthermore, the gut microbiota increases the metabolism of choline, thereby leading to an increase triglycerides level in the liver, due to the lack of VLDL excretion. Furthermore, dysbiosis has been associated with decreased secretion of fasting-induced adipocyte factors [FIAF and angiopoietinlike 4 (ANGPTL4)], which results in the inhibition of endothelial lipoprotein lipase and consequently the lack of hydrolyzation in the liver of triglycerides from VLDL particles. Gluconeogenesis and lipogenesis are upregulated due to an increased number of substrates, particularly short-chain fatty acids, thus promoting hepatic FFAs accumulation by inhibiting AMPK.

  1. When discussing the therapeutic strategies especially in the context of T2DM it is important to address the dietary components (nutrient categories) of the meals and the dietary patterns overall. There is obviously extensive literature on the topic but it would be more appropriate and comprehensive not to limit the approach to the pharmacological one only.  

- Authors: Accordingly, we modified the text, by better raising this point.

The first line approach of the treatment in the context of T2DM is still the modification of lifestyle, which mainly consists in modify some habits.  In most cases, patients who develop T2DM show a sedentary lifestyle and an unbalanced diet, often characterized by a substantial intake of simple carbohydrates and saturated fats. Many authors have extensively discussed this point and demonstrated that mainly Mediterranean, but also vegan and vegetarian dietary patterns should be implemented in public health strategies to obtain a better control glycemic markers in individuals with T2DM. In addition, together with diet, aerobic or resistance exercises training are an essential component of cardiovascular and dysmetabolic diseases prevention and lifestyle intervention programs. Moreover, low-moderate physical activity is crucial to facilitate effectiveness of treatment in order to obtain a good glycemic control.

Reviewer 4 Report

The review discusses associations and mechanism of diabetes and NAFLD with hepatocarcinoma. Treatment options are also discussed. The writing is not always very clear and I think the article needs to be better organized. I would suggest having an opening section on mechanisms by which NAFLD might contribute to diabetes and eventually the carcinoma and then a section on the different treatment options (rather than having multiple sections intertwined).

Lifestyle modifications are mentioned as important; however, these are not discussed in detail. What is the best form of physical activity for these patients? What dietary constituents are recommended? Would pre- or pro-biotics be recommended since gut dysbiosis might contribute to the condition?

Line 24: I don’t think you have defined the abbreviation “NASH”

Lines 35 and 40: It would be of benefit to have references that support the statements on these lines.

Line 40: “nonnegligible” is not a great word to use here...I suggest simply deleting.

Line 50: Please define the abbreviation “NASH”

Lines 63-65: “These criteria will identify a more homogeneous condition than NAFLD, overcoming the difficulties and controversies in defining risky alcohol intake.” I am not sure this sentence makes sense as it implies NAFLD is associated with risky alcohol intake. I suggest re-wording.

Line 87: “Free fatty acids can undergo beta oxidation and TG esterification.” – should this be “Free fatty acids can undergo beta oxidation or TG esterification.”?

Line 89: Please define the abbreviation “FABP” the first time you use it.

Line 106: I suggest deleting the comma in this sentence

Line 139: “Iron2+” – I suggest re-writing this in proper form

Line 147: A reference is needed at the end of the sentence here

Line 154-157: Again, references are needed to support statements on these lines.

Line 162: Please define the acronym / abbreviation “NOX”

Lines 166-168: references are needed to support statements on these lines.

Line 178: Please define the abbreviation “SCFA”

Line 179: Here is it implied that SCFAs promote NAFLD and NAFLD-HCC. Is this true? SCFAs are typically produced by the beneficial gut bacteria and are associated with better metabolic outcomes.

Line 193: Please check that you have defined the abbreviation “SASP”

Line 198: “...depleting the gut microbiota may reduce the development of HCC.” Is this true of all microbiota? Are there some microbiota that may be beneficial in reducing metabolic dysfunction?

Line 210: a reference is needed at the end of this sentence

Line 212: same comment

Line 259: make sure to define the abbreviation “HFD”

Lines 275-280: Please indicate to the reader that the studies referred to here by NCT numbers can be found at www.clinicaltrials.gov.

Line 283: a reference is needed here

Line 288: A reference is needed here for this murine model study

Line 296: a reference is needed here

Lines 332 and 335: should “his expression” be “this expression”?

Line 335: a reference is needed at the end of this sentence on specific miRNAs

Line 339: A reference is needed at the end of this sentence on metabolomics

Line 340: Please make sure you have defined the abbreviation “AFP”

Lines 341-348: References are needed to support these sentences

Line 348: “unbalanced IGF/IGF-1R signaling” – please be more specific here...exactly what is the imbalance?

Lines 350-355: References are needed to support these sentences

Lines 355-360: A reference is needed for this retrospective study

Lines 366-368: A references are needed here

Line 371: “Duration of diabetes before NAFLD diagnosis was associated with a slightly lower risk of HCC” – how do you explain this...it seems contradictory to all the other epidemiological studies mentioned.

Line 377: “FIB-4, NFS, APRI, ELF” – please define these acronyms

Line 380: Provide a reference for the Edinburgh Type 2 Diabetes Study

Line 396: the word “his” is not used properly here. Please chose a different word

Line 396: Here you mention metformin plays a protective role, but earlier in the review article, you stated it was not effective

Line 399: a reference is needed here to the Zhang et al. study

Line 409: a reference is need here

Line 423: Define the abbreviation THCC

Line 428: “CHILD A cirrhosis patients” – what is meant by “CHILD A”?

Lines 442-445: References are needed here

Line 454: “PFS, OS, ORR” – make sure these abbreviations are defined

Line 465: “Chines” – correct to “Chinese”

Line 466: “who randomized” – change to “who were randomized”

Author Response

Dear Rewiever, thank you for your comments that helped us to improve the manuscripte. All suggested corrections have been made in the draft. 

Round 2

Reviewer 1 Report

Where is the figure? I did not see it. 

Author Response

I apologize for the mistake. I upload the figure 1 in the new version of manuscipt.

Reviewer 3 Report

The authors have made a reasonable effort in addressing reviewer's comments. Proofreading for typos is highly recommended the reviewer identified several typos. Moreover, the authors should make sure that their reference list is accurate it terms of the information, accuracy of names and referencing style.

Author Response

Thank you for the suggestion. The reference' style has been revised according to the journal's guidelines. 

Reviewer 4 Report

The authors have adequately revised the manuscript. It needs to be carefully edited for some correction to the English

Author Response

Thank you for the suggestion. The paper has been reviewed by a native English speaker.